# Deep Audio Priors Emerge from Harmonic Convolutional Networks

**Zhoutong Zhang**[1]   **Yunyun Wang**[1,2]   **Chuang Gan**[3]   **Jiajun Wu**[1,4,5]
**Joshua B. Tenenbaum**[1]   **Antonio Torralba**[1]   **William T. Freeman**[1,5]

[1]Massachusetts Institute of Technology   [2]IIIS, Tsinghua University
[3]MIT-IBM Watson Lab   [4]Stanford University   [5]Google Research

## Abstract

Convolutional neural networks (CNNs) excel in image recognition and generation. Among many efforts to explain their effectiveness, experiments show that CNNs carry strong inductive biases that capture natural image priors. Do deep networks also have inductive biases for audio signals? In this paper, we empirically show that current network architectures for audio processing do not show strong evidence in capturing such priors. We propose *Harmonic Convolution*, an operation that helps deep networks model priors in audio signals by explicitly utilizing the harmonic structure. This is done by engineering the kernels to be supported by sets of harmonic series, instead of by local neighborhoods as convolutional kernels. We show that networks using Harmonic Convolution can reliably model audio priors and achieve high performance on unsupervised audio restoration. With Harmonic Convolution, they also achieve better generalization performance for supervised musical source separation. Code and examples are available at our project page: http://dap.csail.mit.edu.

## 1 Introduction

Deep neural networks, in various forms and designs, have been proved extremely successful in both discriminative tasks such as image classification (He et al., 2016), machine translation (Sutskever et al., 2014), speech recognition (Hinton et al., 2012), and in generative tasks such as image and audio generation (Goodfellow et al., 2014; Oord et al., 2017). Recently, Lempitsky et al. (2018) showed that convolutional neural networks (CNNs) come with strong inductive biases suitable for capturing natural image priors. Specifically, a randomly initialized neural network is optimized by fitting a degenerate signal, such as a noisy image, with a random Gaussian noise vector as input. During the fitting process, the network first outputs a clean image, even though it has only seen its noisy version. Lempitsky et al. (2018) argued that this phenomenon shows that CNNs capture image priors by its deep convolutional structure; this is an intriguing perspective on the effectiveness of CNNs in generative modeling of images.

Due to the success of CNNs, audio processing networks usually adopt similar designs from their image processing counterparts, using spatial or temporal convolutions as building blocks. Though such operations could capture image priors, are they capable of capturing audio priors? Is it possible that the task of audio signal modeling needs unique components that cannot be found in image processing networks?

As an attempt to answer the questions above, we investigate whether various audio processing networks can capture audio priors, similar to what CNNs do for images (Lempitsky et al., 2018). Recently, Michelashvili & Wolf (2019) reported that deep priors do exist for Wave-U-Net (Stoller et al., 2018), which can be exploited to perform unsupervised audio denoising. However, the definition of deep priors is slightly different between works by Michelashvili & Wolf (2019) and Lempitsky et al. (2018). Michelashvili & Wolf (2019) reported that the noisy signal causes more violent fluctuations on the spectrogram, which is then utilized as a prior for estimating the noise signal. In contrast, we focus on the deep priors defined by Lempitsky et al. (2018), and investigate whether there is evidence that current deep networks for audio related tasks carry inductive biases for audio priors; if not, what might be an alternative?

In this work, we first empirically show that current architectures for audio signal modeling do not show strong evidence in capturing audio priors. Specifically, we look at two general types of design: temporal CNNs (Michelashvili & Wolf, 2019; Stoller et al., 2018; Aytar et al., 2016) and spectrogram-based CNNs (Shen et al., 2018; Zhao et al., 2018). To test their prior modeling ability, we use the setup identical to Lempitsky et al. (2018): the networks are initialized randomly and tasked to fit a single degenerated audio signal using random Gaussian noises as input. If the network is capable of modeling the signal priors by its structure, it would fit the signal faster than noise. We show an illustrative example in Figure 1, where no substantial evidence is found for temporal or spectral-temporal CNNs, even with this simplest case.

What might be missing? As shown in psychoacoustics experiments, the structure of the harmonic series is closely related to human perception (Moore et al., 1986; Popham et al., 2018). We therefore propose the *Harmonic Convolution*, an operation that explicitly utilizes harmonic structures in audio signals. Then, with multiple experiments, we show that Harmonic Convolution does enable neural networks to model audio signal priors better.

Finally, we show that Harmonic Convolution is useful in downstream applications, and prove its performance by comparing against various baselines. The most natural application is unsupervised audio restoration, where we aim to recover a clean signal from its corrupted version, either by a high power Gaussian noise or aggressive quantizations. In addition, we also demonstrate that networks with Harmonic Convolution achieve better generalization performances for supervised sources separation tasks.

In summary, our contributions are threefold. First, we show that current audio processing architectures do not model audio signal priors naturally. Second, we propose an operation called Harmonic Convolution, serving as an effective inductive bias for neural networks to model audio signal priors. Finally, we demonstrate that networks with Harmonic Convolution achieve state-of-the-art performances in unsupervised audio restoration tasks and improve the generalization ability on supervised musical source separation tasks.

## 2 MOTIVATION

In this section, we first give a brief review of deep image priors. We also provide a short survey on current popular network architecture designs for processing audio signals. Then, we show a motivating toy example, where current architectures fail to model the signal priors, even when the signal is stationary. Finally, we provide a heuristic analysis of why convolution-based approaches may not capture the audio priors, using local signal statistics.

### 2.1 DEEP PRIORS

Lempitsky et al. (2018) first proposed the notion of deep priors on images. Specifically, they show that given a corrupted image $x_0$, a deep neural network $f_\theta$, parameterized by $\theta$, can serve as a natural regularization without any pretaining. Formally, the deep prior method optimizes

$$\min_\theta E(f_\theta(z); x_0)$$

where $E(\cdot; \cdot)$ is a task-specific data term, $x_0$ is the corrupted image, and $z$ is a fixed random noise vector. Lempitsky et al. (2018) showed that optimizing in the parameter space of the neural network $f_\theta$ is surprisingly effective: after several iterations, the optimized network parameters $\theta^*$ gives the restored image $x$ by forwarding the noise vector, i.e. $x = f_{\theta^*}(z)$. Note that the network is randomly initialized and only trained to fit the corrupted signal $x_0$ using a random vector. The fact that it fits the clean signal first suggests that CNNs might be well suited for modeling images, where its structure and operation provide strong inductive biases.

### 2.2 CURRENT NETWORK DESIGNS FOR AUDIO PROCESSING

The network architectures for audio signal processing fall into two broad categories. The first one is to directly apply 1D convolutions on the raw audio signals (Michelashvili & Wolf, 2019; Stoller et al., 2018; Aytar et al., 2016). For instance, Wave-U-Net (Stoller et al., 2018) is a 1D adaptation of the U-Net architecture, which utilizes skip connections and multiscale processings to model signal properties at different time scales. The other category is characterized by performing 2D convolutions on spectrograms (Shen et al., 2018; Zhao et al., 2018). A common practice is first to extract a spectral-temporal representation from the audio signal, then apply 2D convolutions.

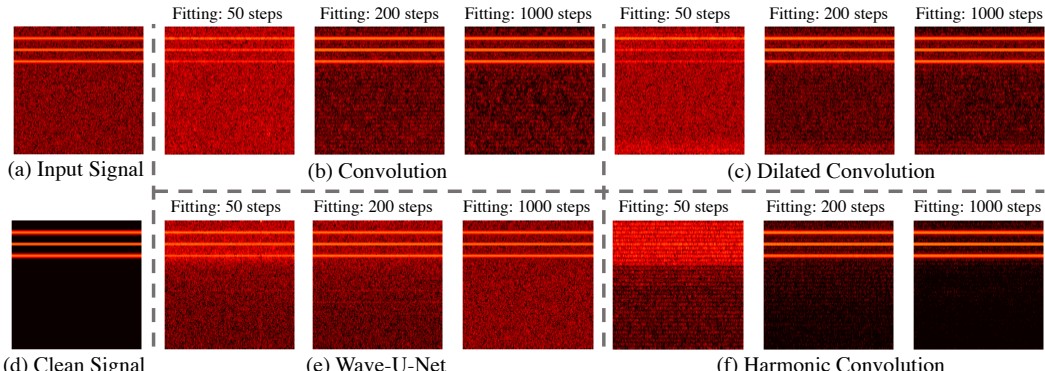

Figure 1: A simple illustrative case: fitting a harmonic sound corrupted by Gaussian noises. The detailed setup of all networks can be found in the appendix. (a) The input to networks is a harmonic sound composed by sinusoids of 1,000, 2,000, and 3,000Hz, plus random Gaussian noises. (d) The clean, target signals. They are shown here as references, not accessible to any models. (b)(c)(e) Both temporal and spectral-temporal convolution networks start with a very noisy output at 50 iterations; they then fit the signal as well as the background noises at 200 iterations. At 1,000 iterations, the output is similar to the one at 200 iterations. This suggest that these networks fit noises and the target signal simultaneously. (f) Harmonic Convolution starts with a noisy output as well, but at 200 steps, the network selectively fits the harmonic series instead. Note that the 1,000-step result is even cleaner than the 200-step one.

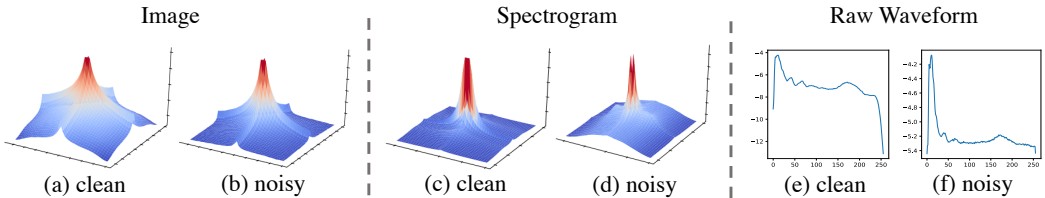

Figure 2: The frequency statistics for images, spectrograms, and raw waveforms. All magnitudes in this figure are in log scale. The DC component is not shown in this visualization. (a) Clean image patch statistics. The $1/f^2$ law can be observed as there is an approximately linear energy falloff from low frequencies to high frequencies in the log scale. (b) Noisy image patch statistics. (c) Clean speech spectrogram statistics. Note that the energy fall-off is different from (a). (d) Noisy speech spectrogram statistics. Note that the energy fall-off along the frequency dimension is different from the fall-off along the temporal dimension, a unique phenomenon to this representation. (e) Clean speech statistics. The power distribution spreads to higher frequencies. (f) Noisy speech statistics.

## 2.3 A MOTIVATING EXAMPLE

Inspired by the deep image prior experiments, we would like to see if the deep architectures above possess the proper inductive biases to model audio signals. To this end, we first test the above architectures to reconstruct a simple signal: a stationary signal composed of 1,000Hz, 2,000Hz, and 3,000Hz sinusoidal waves. The corrupted version of this signal is generated by adding stationary Gaussian noise with a standard deviation of 0.1. As can be seen from Figure 1, all the methods start with very noisy fittings at very early iterations (50 in the figure). Then, they start to fit the signal and the noise at a similar speed, rendering a noisy signal at 200 steps. At 1,000 steps, the network would fit a noisy signal, with slightly less noise than the input signal.

## 2.4 A HEURISTIC ANALYSIS

Here we provide a heuristic analysis on why plain convolution-based networks would fail for modeling audio signal priors by structure. We show that the natural statistics (Torralba & Oliva, 2003) of images and audio signals are different. This suggests that different designs might be needed to capture their priors. In Figure 2, we show, side by side, the frequency statistics of clean natural images, clean speech spectrograms, clean speech signals, and their noisy versions by adding Gaussian noise. All modalities share the same signal-to-noise ratio. The natural images share a $1/f^2$ energy distribution in the frequency domain. This can be seen in Figure 2(a), where the energy falloff is ap-

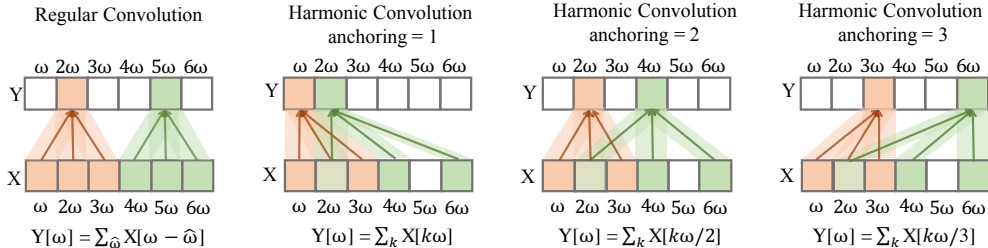

Figure 3: Illustrations for regular convolution and Harmonic Convolution. (a) Regular convolution kernels are supported by a local region. The shape of the support is translational invariant. (b) Harmonic Convolution with an anchoring of 1. This interprets the output frequency as the fundamental frequency for a harmonic series. (c) With an anchoring of 2, the output frequency is the second lower frequency of a harmonic series. (d) Similarly, with an anchoring of 3, the output location is seen as the third harmonics.

proximately linear in the log scale. However, this does not hold for spectrograms or raw waveforms. Details for generating the statistics in Figure 2 can be found in Appendix Section B.

## 3 APPROACH

As shown in Figure 1, the architectures above, despite their success in discriminative tasks, do not show strong evidence for encoding deep audio priors. In light of such facts, we aim to introduce new components for audio signal modeling. We start with harmonic structures, the most common patterns for audio signals, which are also shown to be closely related to human perception (Popham et al., 2018; McPherson & McDermott, 2018). Subsequently, we propose Harmonic Convolution, an operation that treats harmonic structures explicitly.

### 3.1 HARMONIC CONVOLUTION

Harmonic Convolution is designed to explicitly exploit harmonic structures as an inductive bias for auditory signal modeling. Specifically, Harmonic Convolution operates on the discrete Short-Time Fourier Transform (STFT) of a given audio signal, utilizing the spectral-temporal structure of this representation. For audio signal $x[t]$, its discrete STFT spectrogram $X[\omega, \tau]$ is given by

$$X[\omega, \tau] = |\sum_{t=-\infty}^{\infty} x[t]w[t-\tau]e^{-j\omega t}|^2, \tag{1}$$

where $w(\cdot)$ is a locally supported windowing function and $j$ denotes the imaginary unit. Regular 2D convolutions on the spectrogram $X[\omega, \tau]$ with a kernel function $K$ is defined as:

$$(X * K)[\hat{\omega}, \hat{\tau}] = \sum_{\omega=-\Omega}^{\Omega} \sum_{\tau=-T}^{T} X[\hat{\omega} - \omega, \hat{\tau} - \tau]K[\omega, \tau], \tag{2}$$

where the kernel $K$ is supported on $[-\Omega, \Omega] \times [-T, T]$. Note that regular convolution aggregates information in a $2\Omega \times 2T$ window on local regions of $X$. To utilize harmonic structures, we modify this information aggregation scheme to align with harmonics. Specifically, Harmonic Convolution is defined as an operation mapping $X(\omega, \tau)$ to $Y(\hat{\omega}, \hat{\tau})$, where

$$Y(\hat{\omega}, \hat{\tau}) = \sum_{k=1}^{K} \sum_{\tau=-T}^{T} X[k\hat{\omega}, \hat{\tau} - \tau]K[k, \tau]. \tag{3}$$

Note Harmonic Convolution interprets the frequency dimension of the kernel as weights for $K$ harmonic series at each target frequency location $\hat{\omega}$, where regular convolutions interpret the kernel as weights for a local neighborhood at target spectral temporal locations. Figure 3 shows an illustration for both plain convolutions and Harmonic Convolution.

### 3.2 ANCHORS AND MIXING

As indicated by Equation 3, the output at target frequency $\hat{\omega}$ is a weight sum of its $K$ harmonic series, starting from $\hat{\omega}$ to $K\hat{\omega}$. Note that there also exist other possible harmonic series that include $\hat{\omega}$. For example, $\{0.25\hat{\omega}, 0.5\hat{\omega}, 0.75\hat{\omega}, \hat{\omega}, \dots\}$ is also a valid harmonic series, but Equation 3 never

aggregates the information from frequencies lower than $\hat{\omega}$. To over come this problem, we add an extra parameter to Equation 3 called anchor, which indicates the order of harmonics at target frequency location $\hat{\omega}$. Specifically, given the anchoring parameter $n$, we modify Equation 3 as:

$$Y_n[\hat{\omega}, \hat{\tau}] = \sum_{k=1}^{K} \sum_{\tau=-T}^{T} X\left[\frac{k\hat{\omega}}{n}, \hat{\tau} - \tau\right] K[k, \tau]. \tag{4}$$

An illustration for the effect of different anchoring parameters are shown in Figure 3(b)(c)(d). In addition, we can make the output at the frequency location $\hat{\omega}$ depend on multiple anchoring parameters. To this end, we mix different $Y_n$ using a weighted sum: $Y[\hat{\omega}, \hat{\tau}] = \sum_{n=1}^{N} w_n Y_n[\hat{\omega}, \hat{\tau}]$, where $N$ is the largest anchoring parameter, and $w_n$ can be seen as learnable parameters, similar to convolution kernels $K$. Therefore, the final Harmonic Convolution is defined as:

$$Y[\hat{\omega}, \hat{\tau}] = \sum_{n=1}^{N} \sum_{k=1}^{K} \sum_{\tau=-T}^{T} w_n X\left[\frac{k\hat{\omega}}{n}, \hat{\tau} - \tau\right] K[k, \tau], \tag{5}$$

where the learnable parameters are the convolution kernel $K$ and the weights $w_n$.

**Implementation details.** We implement Equation 5 using the Deformable Convolution operation introduced by Dai et al. (2017). For better efficiency, we factorize the 2D kernel $K[k, \tau]$ as the product of two 1D kernels, i.e. $K[k, \tau] = K_f[k]K_t[\tau]$. Anchoring is implemented using grouped Deformable Convolution (Dai et al., 2017), and the weighted sum mixing is implemented as an extra $1\times1$ convolution. Note that the notation is defined on spectrograms, as all operations are implemented by real-valued operations. In experiments, we treat complex STFT coefficients as two separate real-valued channels.

## 4 EXPERIMENTS

In experiments, we test Harmonic Convolution under the deep prior modeling setup introduced by Lempitsky et al. (2018), where the networks are asked to fit a corrupted signal. We define a network's ability to model audio priors as the quality of the audio produced during the fitting process. Under this definition, we show that networks equipped with Harmonic Convolution can model audio priors better than various baselines. As a by-product, Harmonic Convolution performs comparably well with several state-of-art methods for unsupervised audio restoration. Finally, we demonstrate that Harmonic Convolution improves the generalization performance for supervised sound separation.

### 4.1 EXPERIMENT SETUPS

We use the LJ-Speech (Ito, 2017) dataset and the MUSIC (Zhao et al., 2018) dataset. LJ-Speech is a speech dataset consisting of short audio clips of a single speaker reading passages. MUSIC is a video dataset of musical instrument solos crawled from Youtube. We only use their audio tracks for all the experiments.

For fair comparisons, we use the same U-Net (Ronneberger et al., 2015) architecture with different operations, i.e., regular convolutions, dilated convolutions, and the Harmonic Convolution. The details of our network architecture can be found in the appendix. We train all the networks using the Adam optimizer (Kingma & Ba, 2015) with a learning rate of 0.001 for all the experiments.

### 4.2 EXPERIMENTS ON DEEP AUDIO PRIORS

Following Lempitsky et al. (2018), we test networks' ability to model audio priors by fitting a corrupted audio signal using a fixed random input and randomly initialized weights. If the network can produce a restored signal faster and with better quality, then we call this network having a stronger ability to model audio priors.

**Setup.** The random input is drawn from a standard Gaussian distribution, and the weights are initialized by drawing from a zero-mean Gaussian distribution with a standard deviation of 0.02. We test networks that rely on 2D spectral-temporal operations, i.e. regular convolutions, dilated convolutions, and Harmonic Convolution, to fit the complex STFT coefficients of the corrupted signal. We also test networks that operate on the raw waveform, i.e., Wave-U-Net. Again, all operations done by the network are on real numbers; we treat the real and imaginary part of the coefficients as two separate channels. The corrupted signal is generated by adding a clean signal with a zero-mean Gaussian noise with a standard deviation of 0.1. The complex coefficients are generated by taking

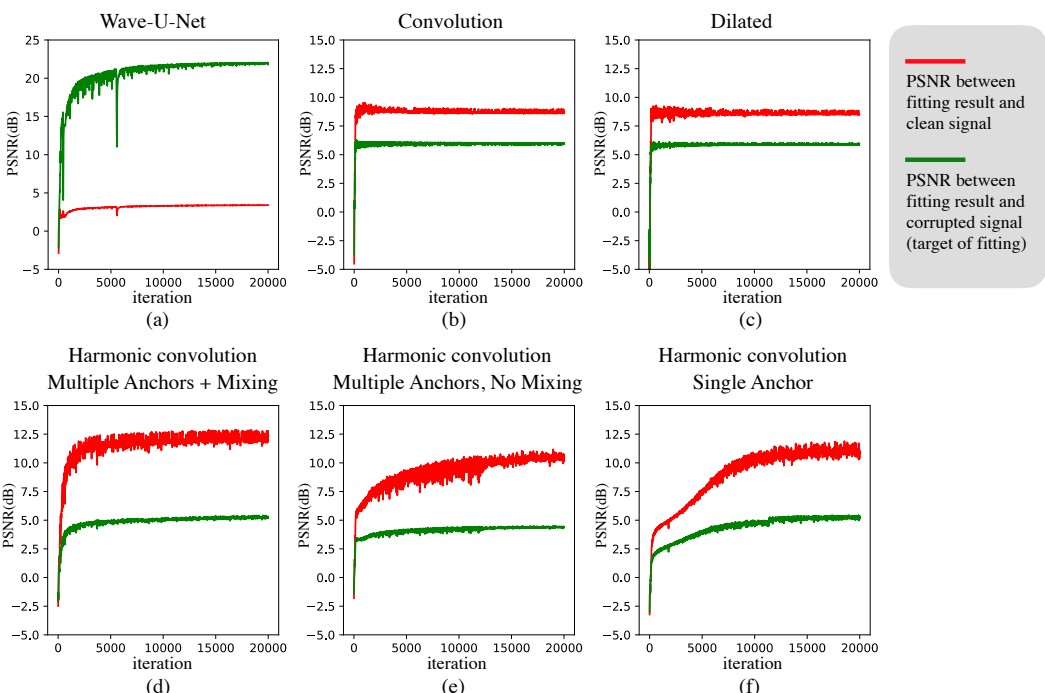

Figure 4: Experiments on deep audio priors using Harmonic Convolution, regular convolution, dilated convolution, and Wave-U-Net. Two PSNR scores are calculated at each iteration: comparing the output against the input noisy signal (green), and against the ground truth clean signal (red). (a) Wave-U-Net fits the input noisy signal fast, but with no strong evidence in producing the clean signal in the process, as the fitting result at each step has a low PSNR against the clean signal. (b)(c) Networks with convolution operations and dilated convolution operations fit the input signal fast, but with only moderate restoration capabilities as measured by PSNR against clean signals. (d) Under the same setup, Harmonic Convolution produces samples of significantly higher quality, compared with (a), (b), and (c). (d)(e)(f) Ablation studies for anchoring and mixing. Without anchoring, the fitting speed measured by restoration quality is slower. With anchoring but without mixing, the final fitting quality is lower.

the Short-Time-Fourier-Transform (STFT) of the corrupted signal, with a box filter of length 1,022. We use a hop length of 64 to provide enough overlapping, reducing the noise introduced by taking the inverse STFT transformation of the fitted result. When comparing against Wave-U-Net, we use the publicly available implementations provided by the authors of Michelashvili & Wolf (2019).

**Results.** We compare the fitting progress of the networks with Harmonic Convolution against various baselines, where the quality of the output signal is measured by Peak Signal-to-Noise Ratio in the temporal domain at each step. As demonstrated in Figure 4, Harmonic Convolution produces a cleaner signal than other methods (-3.5dB boost compared with networks using convolution and dilated convolution). The example of the fitting is randomly sampled from the LJ-Speech dataset. More examples can be found at https://dap.csail.mit.edu.

In addition, we also conduct an ablation study using this fitting process, showing that the design of anchoring and mixing helps with modeling audio priors. As shown in Figure 4 (d) and (f), without the anchoring operation, the fitting speed is much slower. Using anchors without the mixing operation would lead to sub-optimal fitting results.

## 4.3 AUDIO RESTORATION

Similar to the experiments in Lempitsky et al. (2018), networks that model signal priors can be used to perform unsupervised restoration tasks. Here we examine the performance of the network with the Harmonic Convolution on restoring speech and music audios corrupted by random Gaussian noises and aggressive quantizations.

**Setup.** We conducted experiments on both LJ-Speech and MUSIC datasets. For each dataset, we randomly sample 200 audio clips and clip them to 3 seconds long for restoration tasks. For restoring from Gaussian noise, we set the noise to be zero-mean, with a standard deviation of 0.1.

| Methods | Speech | | | | | Music |
|---|---|---|---|---|---|---|
| | CSIG | CBAK | COVL | PESQ | SSNR | SSNR |
| Wiener | 1.00 | 1.37 | 1.00 | 1.08 | 0.01 | 0.68 |
| Wavelet | 1.00 | 1.92 | 1.01 | 1.16 | 2.06 | 6.50 |
| DFL | 1.00 | 1.51 | 1.00 | 1.03 | -0.42 | – |
| DNP+LSA | 1.00 | 1.42 | 1.00 | 1.02 | -3.73 | 4.26 |
| DNP+Wiener | 1.00 | 1.41 | 1.00 | 1.02 | -3.33 | 4.74 |
| Wave-U-Net | 1.00 | 1.36 | 1.00 | 1.02 | -4.62 | 3.99 |
| Regular | 1.17 | 2.15 | 1.10 | 1.09 | 4.13 | 5.07 |
| Dilated | 1.29 | 2.22 | 1.17 | 1.13 | 4.85 | 5.38 |
| Harmonic | **1.76** | **2.36** | **1.43** | **1.20** | **7.12** | **9.85** |

Table 1: Quantitative results on the speech and music restoration tasks.

| Methods | Speech | | | | | |
|---|---|---|---|---|---|---|
| | CSIG | CBAK | COVL | PESQ | SSNR | Human |
| Conv. | 1.00 | 1.03 | 1.00 | 1.09 | 2.78 | 0.095 |
| Dilated | 1.00 | 1.02 | 1.00 | 1.10 | 2.38 | 0.115 |
| Harmonic | 1.00 | 1.01 | 1.00 | 1.09 | 2.05 | **0.79** |

Table 2: Quantitative results on the quantization audio restoration task.

For recovering from quantization noise, we use 1s clips of the 200 randomly sampled speech signals and quantize them into 16 bins, which uniformly covers the range from $-1$ to $1$. The input and output of the network share the same details described in Section 4.2.

**Baselines.** We compare with the following baselines:

- **Wiener:** Wiener filtering (Scalart & Filho, 1996) is an optimization-based methods utilizing a Signal-to-Noise Ratio (SNR) prior. We adopt the implementation where the SNR is estimated from the first 1,024 samples of the signal.

- **Wavelet:** We use the MATLAB implementation for wavelet denoising. Wavelet denoising is based on the sparse prior of audio signals, which assumes the wavelet coefficients should be sparse for clean signals. We use the 8-tap symlet wavelets for this task.

- **DFL:** Deep feature losses (Germain et al., 2018) is a state-of-the-art supervised speech denoising approach using perceptual loss tailored for speech signals. When testing on LJ-Speech with Gaussian noise, we are testing its generalization performances under unseen settings.

- **DNP:** Deep network priors (Michelashvili & Wolf, 2019) is an unsupervised method for audio denoising using deep priors. Contrary to our method, the authors observed that during the fitting process, the injected noise varies more violently than the signal itself. Therefore, this property can be used to identify noise regions and provide an SNR estimate for traditional filtering methods such as LSA (Ephraim & Malah, 1985) or Wiener filtering (Scalart & Filho, 1996).

**Metrics.** For the speech restoration task, we adopt multiple quality metrics to measure the audio restoration results, including the mean opinion score (MOS) predictor of signal distortion (CSIG), the MOS predictor of background-noise intrusiveness (CBAK), the MOS predictor of overall signal quality (COVL), the perceptual evaluation of speech quality (PESQ), and the segmental Signal-to-Noise Ratio (SSNR). For the music restoration task, we only report results measured by the SSNR, since other metrics do not apply to the music signals. CSIG, CBAK, COVL, and PESQ are on a scale of 1 to 5, where 5 is the best quality.

**Results.** Table 1 summarizes our results against the mentioned baselines over these metrics. Our method consistently outperforms all the baselines according to all measures by a considerable margin. This directly demonstrates that Harmonic Convolution can make neural networks more suited for modeling audio signals.

We also conduct an experiment to verify that Harmonic Convolution is not limited to the additive Gaussian noise case. In this experiment, we quantize 200 randomly sampled one-second speech signals into 16 bins, uniformly covering the range of $[-1, 1]$. The results are reported in Table 2. Since the scores for each metric are rather close, we conduct a carefully designed perceptual experiment with human listeners. For each model, we take 200 audio restorations from the same set for evaluation and each audio is evaluated by three independent Amazon Mechanical Turk (AMT) workers. We present the model results in random order and ask the annotator to select the audio clip that has the best restoration results. The results show nearly 80% of the subjects vote for the results produced by the network using Harmonic Convolution.

| Methods | Guitar | | | Xylophone | | | Congas | | |
|---------|------|------|------|------|------|------|------|------|------|
| | SAR | SIR | SDR | SAR | SIR | SDR | SAR | SIR | SDR |
| Conv. | 13.3 | 6.1 | 4.6 | **14.0** | 10.3 | 7.8 | 12.7 | 6.6 | 4.9 |
| Dilated | 14.6 | 7.0 | 5.7 | 13.8 | 12.6 | 9.2 | 12.8 | **7.8** | 6.0 |
| Harmonic | **15.1** | **7.9** | **6.7** | 13.7 | **14.0** | 9.9 | **13.2** | 7.6 | **6.1** |

Table 3: Quantitative results on the generalized sound separation task. The units are in dB.

## 4.4 GENERALIZED SOUND SEPARATION

Here we examine whether the Harmonic Convolution can improve generalizations of the supervised sound separation task, compared with regular and dilated convolutions.

**Setup.** To evaluate the generalization ability of sound separation networks, we select five musical instruments from the MUSIC dataset (Zhao et al., 2018): violin, cello, congas, erhu, and xylophone. Each category consists of 50 6-seconds solo audio clips. We aim to test the model's ability to generalize to unseen musical instrument mixtures. Specifically, the model is tasked with separating the sound of a target instrument from a clip that also contains sounds of another instrument. During training, we avoid using clips that contain sounds of a selected holdout instrument class, so that the model has never 'heard' of the sound of that type of instrument before. We then test the separation performance of this model on mixtures made from the sounds of the model's target instrument and the holdout instrument. In particular, we train models to separate the sound of violins and use three different holdout instruments to test its generalizability (congas, violin, and xylophone). The input to the network is the complex STFT coefficients of the mixed audio, using a box filter of size 1022 and a hop length of 64. The output of the network is a ratio mask, calculated as the ratio of the spectrogram between the sound of the target instrument and the input mixed sound. To produce separated audios, we apply an inverse STFT transformation to the input complex STFT coefficients multiplied by the predicted ratio mask.

**Implementations.** We adopt the Mix-and-Separate framework (Zhao et al., 2018) for this task. We first generate a synthetic separation training set by mixing the audio signals from two different audio clips and then trains a neural network to separate the sound of the target instrument, e.g., violin.

During training, we take a 1-second mixed audio clip as input and transform it into complex coefficients using the Short-Time-Fourier-Transform (STFT). The spectrogram is then fed into a U-Net, whose architecture is described in Section 4.1. The U-Net outputs a ratio mask, and the network is trained by minimizing the L1 loss between the predicted ratio masks and the ground-truth masks. We use a 90:10 train-val split, and test the performance on the mixture between sounds of the target instrument and the sounds of the holdout instrument.

**Results.** We compare the performance of the proposed Harmonic Convolution against the regular and dilated convolutions used in previous papers. We use the Signal-to-Distortion Ratio (SDR), Signal-to-Interference Ratio (SIR), and Signal-to-Artifact Ratio (SAR). metrics from the open-source `mir_eval` library (Raffel et al., 2014) to quantify performances. Quantitative results are reported in Table 3. We observe that while all networks suffer when tested on mixtures under novel recording conditions, the Harmonic Convolution exhibits better generalization performances. This suggests that Harmonic Convolution not only can be used as a prior for unsupervised tasks but also has the potential to be helpful for supervised tasks.

## 5 RELATED WORK

**Deep priors.** Our work is inspired by the recent paper on deep image priors (Lempitsky et al., 2018), which shows that the structure of CNNs imposes a strong prior to restore a single original image from the degraded image. The idea of deep priors has also shown to be useful in many applications, including semantic photo manipulation (Bau et al., 2019), image super-resolution (Shocher et al., 2018), and image decomposition (Gandelsman et al., 2018). While most prior papers focused on images, little work has explored deep priors on audio signals.

**Deep learning for auditory signal modeling.** Deep networks have gained remarkable success on the audio signal modeling, such as speech recognition (Hinton et al., 2012; Amodei et al., 2016), sound separation (Stoller et al., 2018; Zhao et al., 2018; 2019), audio denoising (Rethage et al., 2018; Germain et al., 2018), audio generation (Oord et al., 2017; Mehri et al., 2017), text to speech synthesis (Wang et al., 2017; Shen et al., 2018), and voice conversion (Hsu et al., 2017). A detailed

survey can be found at (Purwins et al., 2019; Qian et al., 2019). However, it remains unclear if these architectures themselves capture the audio signal priors. Most related to our work is Michelashvili & Wolf (2019), where they used deep networks as a prior to estimate the SNR prior on the spectrogram and then used classical post-processing algorithms to perform the speech denoising. Note that the deep prior mentioned in works of Mehri et al. (2017) is different from the ones mentioned in Lempitsky et al. (2018). The former uses fitting time variances as noise indicating priors. We aim to find designs that bias towards clean audio signal.

**Structured operators for deep models.** Various algorithms have been proposed to include more dynamic structures beyond translational invariant kernels. Dai et al. (2017) proposed a convolution operator with dynamic offsets at each spatial location, which improves the object detection accuracy by distinguishing foreground and background features. Shelhamer et al. (2019) introduced a spatially adaptive Gaussian kernel to help scale and steer convolution kernels, without introducing a significant amount of free parameters. Ravanelli & Bengio (2018); Sainath et al. (2013) proposed to learn parametrized filter banks that adapt to the convolution structure. While most of these algorithms focus on the discriminative tasks, our research topic is more related to generative tasks. Moreover, the learned filter banks in Ravanelli & Bengio (2018); Sainath et al. (2013) are used only in the first layer and not invertible in general, making them less suitable for the generative tasks where the output is an audio signal.

**Psychoacoustics.** Harmonic structures are closely related to the human perception of audio signals. The famous missing fundamental auditory illusion suggests that human can infer the missing fundamental frequency by only hearing its overtones (Todd & Loy, 1991). Moore et al. (1986) showed that shifts in harmonic components would be perceived as separate tones. More recently, Popham et al. (2018) showed that the harmonic structure plays an important role for human to solve the cock-tail problem, where inharmonicity would cause difficulties for human to track speakers for the cocktail party problem. McPherson & McDermott (2018) showed that pitch perception is closely related to the harmonicity of the sound.

## 6 CONCLUSION

In this paper, we examined various architectures on deep audio prior modeling. We then proposed a novel operation called Harmonic Convolution, which can help networks better capture priors in audio signals. We showed that fitting a randomly-initialized network equipped with Harmonic Convolution can achieve high performance for unsupervised audio restoration tasks. We also showed that Harmonic Convolution improves the generalization ability in sound separation.

**Acknowledgment.** We would like to thank Josh H. McDermott for helpful discussions. This work is supported by the NSF award #1447476 and IBM Research.

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

## A  NETWORK ARCHITECTURE

The U-Net used by all methods consists of 5 blocks, each block contains two operation layers, which can be instantiated by regular convolutions, dilated convolutions or the Harmonic Convolution. The feature map is downsampled by average pooling with a stride of 2 and a window size of 2. Down-sampling is performed after the first two operation layers. As in common designs of U-Net, the upsampling is performed before the last two layers through bilinear upsampling with a scale of 2. Finally, we attach a $1{\times}1$ convolution layer after the U-Net's last layer to give the final output. Feature map sizes for all the layers are [input→input, input→35], [35→35, 35→70], [70→70, 70→70], [140→140, 140→35], [70→70, 70→35], then followed by a $1{\times}1$ regular convolution layer mapping the final 35 channels to the desired number of output channels.

In addition, we also keep the kernel sizes fixed for all different operations. For regular convolutions, we use $7{\times}7$ kernels for all the layers. For dilated convolutions, we use the same kernel size ($7{\times}7$) with dilation of 3 for all the layers. For the Harmonic Convolution, we use a frequency kernel $K_f$ of length 7 and a temporal kernel $K_t$ also of length 7. We use 7 anchors ($N{=}7$ in Equation 5) for all the Harmonic Convolution operations. We use instance normalization (Ulyanov et al., 2016) and ReLu activations (Krizhevsky et al., 2012) for all the experiments.

## B  DETAILS FOR NATURAL STATISTICS ANALYSIS

The image statistics are computed on 1,000 images randomly sampled from ImageNet (Russakovsky et al., 2015). The audio spectrogram statistics are computed on 1,000 speech signals randomly sampled from the LJ-Speech dataset (Ito, 2017). We compute their spectrograms using the Short-Time-Fourier-Transform and calculate the spatial frequency distribution of the spectrograms as if they are images. The 1D frequency distribution of the audio signals is calculated on the same speech signals, where we interpret audio clips like 1D image patches.

## C  RELATING HARMONIC CONVOLUTION WITH DILATED CONVOLUTION

The simplest form of Harmonic Convolution is defined in Equation 3:

$$Y(\hat{\omega}, \hat{\tau}) = \sum_{k=1}^{K} \sum_{\tau=-T}^{T} X[k\hat{\omega}, \hat{\tau} - \tau] K[k, \tau]. \tag{6}$$

Contrasting dilated convolution, one can write its operation as

$$Y(\hat{\omega}, \hat{\tau}) = \sum_{\omega=-\Omega}^{\Omega} \sum_{\tau=-T}^{T} X[\hat{\omega} - \omega * n, \hat{\tau} - \tau] K[k, \tau], \tag{7}$$

where n is the dilation parameter.

