# OpenReview forum: "Deep Audio Priors Emerge From Harmonic Convolutional Networks"
_ICLR.cc/2020/Conference — Accept (Poster)_

### Official Review · AnonReviewer2 · 2019-10-18
**Official Blind Review #2**

**Rating:** 6

**Review:**

In this paper, the authors introduce a new convolution-like operation, called a Harmonic Convolution, which operates on the STFT of an audio signal.  This Harmonic convolution are like a weighted combination of dilated convolutions with different dilation factors/anchors.  The authors show that for noisy audio signals, randomly initialized/untrained U-Nets with harmonic convolutions can yield cleaner recovered audio signals than U-Nets with plain convolutions or dilated convolutions.  The authors beat a variety of audio denoising tasks on a variety of metrics for speech and music signals.  The authors also show that harmonic convolutions in U-Nets are better than plain and dilated convolutions in U-Nets for a particular sound separation task.

I recommend a weak accept for this paper because a new architecture for audio priors was presented, with reasonable empirical data supporting that this architectural choice an improvement over other more immediate alternatives.  It is important to extend the work on deep nets for imaging to other domains, such as audio. My recommendation is not stronger because of the following concerns.

I think the paper could be strengthened by
(a) a comparison to other methods (outside the current framework) for sound separation
(b) a significant clarification of Figure 4.  The authors claim that this data shows that Harmonic Convolutions produce a "cleaner signal faster" than other methods.  When I look at Figure 4abcd, it appears that the Convolution and Dilated Convolutions fit a clean signal faster (it is just not as clean.  Additionally, the Wave-U-Net appears to reach the same accuracy as the Harmonic Convolution with many fewer iterations (while also continuing to get much higher PSNRs).  Perhaps I am misreading this plot, but it is not obvious to me that this plot supports the claims the authors are making.
(c) The authors should present what they mean by a dilated convolution using the notation of the paper.
(d) In Figure 2, it is unclear to me how the 1/f^2 law is observed in (a) but not in (c) or (e).

**Experience Assessment:**

I have published one or two papers in this area.

**Review Assessment: Checking Correctness Of Derivations And Theory:**

N/A

**Review Assessment: Checking Correctness Of Experiments:**

I assessed the sensibility of the experiments.

**Review Assessment: Thoroughness In Paper Reading:**

N/A

---

> ### Author Response · Authors · 2019-11-10
> **Response to Reviewer #2's comments**
>
> Thank you for your constructive comments! We would like to address your concerns as follows:
>
> 1. Clarification on Fig. 4.
>
> We rewrote the caption for Fig. 4. Specifically, for Wave-U-Net, the green curve indicates the fitting result compared against the noisy target, and the red curve is the result evaluated against the clean signal. Therefore, Wave-U-Net fits the noisy target fast but does not produce the clean version of the signal during fitting. For Convolution and Dilated Convolution networks, they do fit faster but saturates with low-quality output. Harmonic Convolution produces much better results, which is ~3.5 dB higher. We highly recommend listening to examples at https://anyms-sbms.github.io to feel the difference.
>
> 2. Dilated convolution in paper’s notation.
>
> We have added a section in the appendix to include dilated convolution in the paper's formulation.
>
> 3. Clarification on Fig. 2.
>
> Since the plots in Fig. 2 are log-scale, one would expect nearly linear fall-off of energy from low-frequency components to high-frequency components, which is the case of (a). But (c)(e) exhibit drastically different fall-offs of energies compared with (a). We have modified the caption of Fig. 2 to be more specific.
>
>
> We compared our model with unsupervised/supervised NMF for sound source separation, a common unsupervised baseline for this task. The evaluations are reported as follows:
> ----unsupervised----
> guitar:          SDR: 2.17   SIR: 2.78   SAR: 14.19
> congas:        SDR: -0.20  SIR: 0.23   SAR: 14.76
> xylophone:  SDR: 2.04   SIR: 3.61   SAR: 12.13
>
> ----supervised----
> guitar:          SDR: 5.97   SIR: 7.56   SAR: 12.81
> congas:        SDR: 1.77  SIR: 2.76   SAR: 11.97
> xylophone:  SDR: 8.08   SIR: 12.33   SAR: 11.72
>
> Please let us know for any questions. Thanks again for your suggestions, which have made this submission stronger.
>
> Thanks,
> Authors

---

### Official Review · AnonReviewer3 · 2019-10-22
**Official Blind Review #3**

**Rating:** 6

**Review:**

This paper studies the problem of how to design generative networks for auditory signals in order to capture natural signal priors. Compared to state-of-art methods in images [Lempitsky et al., 2018], this problem is not so easy on audio signals. Existing work [Michelashvili &Wolf] trains generative networks to model signal-to-noise ratio rather than the signal itself. This paper proposes a new convolutional operator called Harmonic Convolution to improve these generative networks to model both signals or signal-to-noise ratio. Applications on audio restoration and source separation are given.

The paper starts to show that an existing generative network Wave-U-Net does not capture audio signal priors. The explanation in Fig 2 on why this is the case seem to me not so clear. Are you trying to show that the Wave-U-Net does not work since there is no 1/f^2 law for clean audio signals?

The Harmonic Convolution is similar to deformable convolutions, but specifically designed to capture audio harmonics. It is further combined with the idea of anchors and mixing to capture fractional frequencies. The explanation of this section is slightly unclear. There is a little typo in Formula 1 for the STFT spectrogram, I would use the modulus |.| rather than || . ||. Is Harmonic Convolution applicable to complex STFT coefficients as well? It seems to be yes based on Section 4.2. If so it would be better to define the operator in a more general notation.

Numerical experiments show that the Harmonic Convolution improves over existing regular and dilated convolutions in various settings. Section 4.2 aims to fit the complex STFT coefficients of corrupted signals. However, the setting is less clear to me for both the unsupervised speech/music restoration and supervised source separation problems. In Section 4.3 and 4.4, is the x_0 (defined in Section 2.1) complex-valued STFT coefficients or something else? It seems to me x_0 = ratio mask in Section 4.4. What is the L1 loss defined in Section 4.4? To obtain the final separated audio waveform, an inverse STFT is applied on what? These details can be written in supplementary material if more space is needed. After all, the numerical results seem to me encouraging.

**Experience Assessment:**

I have read many papers in this area.

**Review Assessment: Checking Correctness Of Derivations And Theory:**

N/A

**Review Assessment: Checking Correctness Of Experiments:**

I carefully checked the experiments.

**Review Assessment: Thoroughness In Paper Reading:**

I read the paper at least twice and used my best judgement in assessing the paper.

---

> ### Author Response · Authors · 2019-11-10
> **Response to Reviewer #3's comments**
>
> Thank you for your helpful suggestions and we would like to address your concerns as follows:
>
> 1. Better explanatory texts for natural statistics comparison.
>
> We have modified the caption for Fig. 2 and text in Sec 2.4 to be more clear about the natural statistics analysis. This analysis is intended to contrast the natural statistical differences among the representations, to indicate that different modeling approaches are needed for each of them. Models that capture image priors well might not transfer to spectrograms or raw waveforms.
>
> 2. Equation 1 typo fixed.
>
> 3. Complex Coefficient vs Spectrograms.
>
> Thanks for the suggestion. We intentionally use the spectrogram notation as we do not use complex-valued kernels with complex-valued convolution. Yet in order to generate the audio signal, we simply generate the real and imaginary parts of the STFT coefficients such that we can convert them to waveform using inverse STFT.
>
> We have modified the text in the implementation details in Sec. 3 and the setup paragraphs in Sec. 4.2, 4.3, and 4.4 to make this point.
>
> 4. Details in the experiments to clear up the settings.
>
> We have modified the text in Sec. 4.2, 4.3, and 4.4 to make the details more clear.
>
> For experiments in Sec. 4.2 and 4.3, the network’s output is the complex STFT coefficient, the raw waveform is then recovered by inverse STFT using the overlap-and-add method. For experiments in Sec 4.4, the output of the network is the ratio mask, and the separated audio is generated by an Inverse STFT operated on the input STFT coefficients multiplied by the predicted ratio mask. The L1 loss is calculated between the predicted ratio mask and the ground truth ratio mask.
>
> Please let us know for any questions. Thanks again for your suggestions, which have made this submission stronger.
>
> Thanks,
> Authors

---

### Official Review · AnonReviewer1 · 2019-10-23
**Official Blind Review #1**

**Rating:** 6

**Review:**

The paper considers the effectiveness of standard convolutional blocks for modelling learning tasks with audio signals. The effectiveness of a neural network architecture is assessed by evaluating its ability to map a random vector to a signal corrupted with an additive noise. Figure 1 illustrates this process with a network taking a single standard normal vector as input and having a single target output consisting of some signal corrupted with additive noise.

The paper is not well written and it is rather difficult to follow. It is also not well structured with a number of relevant concepts properly described only sections after they appear for the first time.

The first issue I had with the paper was the notion of audio prior. It was only after reading the whole paper that I have realized what this means. Having said this, it is unclear why the employed notion would work in general. I see why it could work when the distribution of the input vector and additive noise are correlated. This has not been clarified nor discussed and I believe it merits a couple of sentences.

In the introduction, the paper states "... unlike CNNs for image modelling, the design of deep neural networks for auditory signals has not yer converged". First, it is not clear what it means for the architecture to converge. If we assume that it refers to standard convolutions with a couple of widely accepted filter size and max pooling, the I would say that in speech recognition the structures that work are quite similar for mel-frequency coefficients or fbank features as inputs (which are again convolutional feature extraction layers).
Shortly after this, there is a question on justification of network designs. I disagree with a potential implication that this is well understood for image processing. For some insights relevant to speech, the work by Mallat ("Group invariant scattering", 2012) might be useful.

Figure 1 and the paragraph just below its caption are not clear. It is not explained what is the input/output of the network and this is of great importance for the understanding of the illustration in Figure 1.

The introduction does not explicitly define the notion of audio prior and the whole paper is about this. In my opinion, it is wrong to assume that a reader has seen the paper by Lempitsky et al (2018).

Section 2.1, the optimization objective as formulated implies that z and x_0 are completely independent. I do not see how any meaningful conclusion can be derived by fitting a map between independent input and output vectors. Some assumption is required for the proper notion of "audio prior" (if not, then a discussion arguing for the opposite).

Section 3, opening paragraph concludes that standard CNNs are not the best blocks to model learning tasks with audio signals. For this implication, one needs the exact structure of CNN network and more details with regard to the experiment itself. In particular, there are deep CNNs (with mel-frequency coefficients as inputs) that work very well in speech recognition (e.g., on noisy datasets such as aurora4). This illustration does not say anything about the influence of the depth and number of convolutional blocks on a learning task. The language should be more moderate here and, in general, some additional work is required on the motivation of harmonic convolutions.

In my understanding, harmonic convolutions are a special case of deformable convolutions (Dai et al., 2017). In essence, standard convolution is applied over time and deformable over the frequency axis of a spectrogram. The main contribution seems to be in that the work provides a structure to the offsets in Dai et al. (Section 2.1, 2017). If I am correct, then this should be discussed in details and the harmonic convolution needs to be placed in the context of prior work. It might help by starting with a review of that work and then introducing the imposed structure on the offset vectors. I am having problems understanding the illustration in Figure 3.

In the experiments, the work is evaluated on signal de-noising (audio restoration) and sound separation.

The first task is carried out under the assumption that the signal has been corrupted with Gaussian noise and shows advantages of the approach over baselines which include standard convolutional networks. It would be interesting here to see how the depth of a convolutional network affects the performance. Also, as the approach is (in my understanding) a special case of deformable convolutions it would be insightful to show an experiment with that baseline. While additive noise is difficult on its own, many signals are corrupted by channel noise. It would be interesting to add an experiment with different types of channel noise and which network design is more likely to de-convolve the noise from the signal.

The second experiment deals with separation of sounds of different musical instruments and the results again show advantages of harmonic convolutions over the baselines.


**Experience Assessment:**

I do not know much about this area.

**Review Assessment: Checking Correctness Of Derivations And Theory:**

N/A

**Review Assessment: Checking Correctness Of Experiments:**

I assessed the sensibility of the experiments.

**Review Assessment: Thoroughness In Paper Reading:**

I made a quick assessment of this paper.

---

> ### Author Response · Authors · 2019-11-10
> **Response to Reviewer#1's comments**
>
> Thank you for your constructive comments about our manuscript. We have modified the paper in the following way and hopefully, this could address your concerns:
>
> 1. Introducing Harmonic Convolution with the context of deformable convolution
>
> We emphasize that our method is only loosely related to deformable convolution. In our paper, we show that the harmonic structure is important for the network to capture priors in audio signals. This structure is general and does not need to be learned. In contrast, deformable convolution emphasizes that learning custom offsets for convolutional kernels can boost object detection performance. The only connection we have now with deformable convolution is that we used their implementation of fractional bilinear sampling during convolution as a building block for our method. Such an implementation can and will be replaced for better efficiency. To clarify this explicitly, in revision, we have added a section in related work to discuss the various structured operators in deep learning.
>
> We agree that it’d still be a good idea to compare with deformable convolution. We’ll include the results in a later revision by Nov 15.
>
> 2. An early and formal notion of Deep Priors, without assuming moderate exposure to Lempitsky et al. (2018):
>
> We have modified the first paragraph in the introduction section, adding a highly summarized overview of Lempitsky et al. (2018), to establish the foundation of deep priors.
>
> 3. More accurate narratives towards deep learning models in image and audio processing:
>
> We modified it to convey the core message and motivation of this paper: investigating whether various designs for generative audio signal modeling could capture audio priors by their structure. We do recognize that CNNs prove effective for various discriminative tasks, yet our task is more related to generative modeling of audio signals.
>
> 4. A more precise description of Fig. 1:
>
> We’ve added more descriptions in the caption of Fig. 1 to explain the setup.
>
> 5. Relate Sec. 2.1 to audio priors, explain why mapping random vector z to corrupted target signal is meaningful and why this matters.
>
> As demonstrated in Lempitsky et al. (2018), when the neural network is fitting a corrupted signal x_0 with randomly initialized weights and with random vector z as input, it would first learn a mapping from z to the clean version of $x_0$. The argument being that the network provides an implicit regularization, where the clean signal is much easier for it to fit. Thus, the inductive bias implied by the network itself can be seen as more suitable for modeling images. In terms of the audio signals, we aim to use the same setup to probe if the inductive biases implied by various models are suited for audio signal modeling.
> We modified Sec. 2.1 by adding more explanation at the end to make this point more clear and accessible. We also added the missing definition of $x_0$ in the text.
>
> 6. The opening sentence in Sec. 3, more on the motivation of harmonic convolution:
>
> The opening of Sec. 3 is not about whether CNNs are the best blocks for learning from audio signals (in fact, we acknowledged their success in supervised learning tasks), but to state that they do not, by nature, capture audio priors as shown in Figure 1 and our experiments in general. We clarify that whether a model can capture audio priors is not necessarily related to their performance in supervised learning tasks such as speech recognition. We have edited this sentence to be more explicit.
> In light of the facts above and motivated by experiments in psychoacoustics, we aim to exploit the harmonic structure in audio signals explicitly, which leads to the design of harmonic convolutions.
> The motivation above was added to the paper, as suggested.
> As for the illustrations, we are assuming this refers to Fig. 1. We added text referring to the exact architecture of the models, which is described in the appendix.
>
> 7. Confusion about Fig. 3
>
> We have revised Fig. 3 to be more straightforward. We added the annotation for input and output, as well as equations to make the anchoring parameter explicit. We also changed the aggregation annotation to be more clear.
>
> 8. Different types of noise
>
> Thanks for the suggestion. We focus on additive Gaussian noise in our experiments because that is the most common approximation of the channel noise during transmission. We also showed restoration results under very aggressive quantization, which is rather signal-dependent instead of additive. We are happy to experiment with other types of channel noise if there is any specific suggestion.
>
> We are working on the following experiments to address the points you raised:
>
> 1. Comparing denoising results with Deformable Convolution.
>
> 2. Varying the depth of each model and report performance change, in the context of denoising.
>
> Please let us know for any questions. Thanks again for your suggestions, which have made this submission stronger.
>
> Thanks,
> Authors.

---

> > ### Author Response · Authors · 2019-11-15
> > **Results for Additional Experiments**
> >
> > 1. Comparing denoising results with Deformable Convolution:
> > We report the denoising performance of Deformable Convolution on the LJ-Speech dataset. We use the same network structure and hyper-parameter described in the paper. To keep the number of parameters consistent, we do not use extra convolution layers to predict kernel offsets, as described in the original deformable convolution paper. We treat the offsets as parameters and optimize them together with the network's weight.
> >
> > Deformable Convolution:
> > CSIG:  1.00111 CBAK:1.66844   COVL:1.00326   PESQ:1.05002  SSNR:-1.52444
> >
> >
> > 2. Varying Depth:
> > We also experimented with varying depth of the network and test its performance. To keep the experiments consistent, we use the same hyper-parameters as described in the paper, including the learning rate.
> >
> > The channels for adding layers are:
> >  [input→input, input→35], [35→35, 35→70],[70→70, 70→140], [140→140, 140→140], [280→280, 280→70], [140→140, 140→35], [70→70, 70→35]
> > The corresponding metrics are:
> > CSIG: 1.00197   CBAK:1.67211   COVL:1.00507    PESQ:1.05218    SSNR:-1.48005
> >
> > The channels for removing layers are:
> >  [input→input, input→35], [35→35, 35→35],[70→70, 70→35]
> > The corresponding metrics are:
> > CSIG: 1.00119    CBAK:1.67021   COVL:1.00383    PESQ:1.05173    SSNR:-1.50226
> >
> > We also updated the corresponding results on our webpage:
> > https://anyms-sbms.github.io/speech_denoising.html
> >
> > The shallow version of our network is still capable of generating restored content to some degree, yet the deep version seems to overfit to the noise.

---

### Public Comment · ~Joe_Renner1 · 2019-10-18
**Implementation Question**

Hello,

Great paper! One quick question:

In the case where the anchor value n is > 1, how do you aggregate lower order harmonics if the target frequency location is not divisible by 2 ^ (n - 1)?

For example, the target frequency bin is 5, and the anchor is 2, so the first lower order harmonic frequency bin in the kernel would be computed as 5/2.

You mention you used deformable convolution to implement equation 5, which would lead me to believe you used bilinear interpolation to compute the value if the offset is fractional, but I just wanted to make sure I was understanding this correctly.

Thanks!

---

> ### Author Response · Authors · 2019-10-18
> **Re: Implementation Question**
>
> Hi, Joe
>
> Thank you for your comment!
>
> Yes your understanding is correct. We use the bilinear interpolation for sampling the fractional frequencies.

---

> > ### Public Comment · ~Hirotoshi_Takeuchi1 · 2020-01-07
> > **Similar implementation question**
> >
> > Hello.
> >
> > Your paper interested me!
> >
> > I have a similar question.
> > I want to know how to calculate X[kω/n,t] when Ω < kω/n.
> >
> > Thanks.

---

> > > ### Author Response · Authors · 2020-01-08
> > > **Re: Similar implementation question**
> > >
> > > Hi,
> > >
> > > We use zero paddings for all the operations.
> > >
> > > Thanks,
> > > Authors

---

### Author Response · Authors · 2019-11-10
**General Response to Reviewers' Comments**

We thank the reviewers for their efforts and helpful comments. We have revised the paper to address concerns on the presentation and are running additional experiments as suggested. We will update the paper again before Nov 15 to include the results of those experiments.

Text revisions include (highlighted in the revised manuscript):

1. A better introduction of deep priors (R1):

We have modified the first paragraph in the introduction section, adding a highly summarized overview of Lempitsky et al. (2018), to establish the foundation of deep priors.

2. More accurate arguments on deep models for image and audio processing (R1):

We modified Sec. 2 to convey the core message and motivation of this paper: investigating whether various designs for generative audio signal modeling could capture audio priors by their structure. We do recognize that CNNs prove effective for various discriminative tasks, yet our task is more related to generative modeling of audio signals.

3. Clearer introduction of Fig. 1 (R1):

We’ve added more descriptions in the caption of Fig. 1 to clarify the setup.

4. Better relate and motivate the design of Harmonic Convolution in Sec. 2.1 (R1):

We modified Sec. 2.1 by adding more explanation at the end to make it more clear and accessible. We also added the missing definition of x_0 in the text.

5. Improve Sec. 3 to be more accurate on CNN models for image and audio processing (R1):

We modified this sentence to be more explicit, acknowledging their success in discriminative tasks.

6. Add deformable convolution into context (R1):

We added a section in the related works to discuss the structured operators in various deep learning models.

7. Better explanation to Fig. 2 (R2, R3):

We have modified the caption for Fig. 2 and text in Sec. 2.4 to be more clear about natural statistics analysis.

8. More clarification on Harmonic Conv in Approach section (R3):

We have modified the text in the implementation details in Sec. 3 and the setup paragraphs in Sec. 4.2, 4.3, and 4.4 to make this point.

9. Better Explanation of Fig. 4 (R2):

We rewrote the caption for Fig. 4 to make it more clear.

10. Dilated Convolution in the paper’s notion (R2):

We have added a section in the appendix to include dilated convolution in the paper's formulation.

11. Typo in Equation 1 (R3):

We fixed the typo indicated by R3.

Extra experiments include

1. Compare against deformable convolution (R1)

We report the speech denoising result using deformable convolution:
Deformable Convolution:
CSIG:  1.00111 CBAK:1.66844   COVL:1.00326   PESQ:1.05002  SSNR:-1.52444

2. Performance of models with varying depth (R1)
We added extra layers to our current architecture. The corresponding metrics in speech denoising are:
CSIG: 1.00197   CBAK:1.67211   COVL:1.00507    PESQ:1.05218    SSNR:-1.48005

We also removed the middle layers and the corresponding metrics in speech denoising are:
CSIG: 1.00119    CBAK:1.67021   COVL:1.00383    PESQ:1.05173    SSNR:-1.50226

We also updated the corresponding results on our webpage:
https://anyms-sbms.github.io/speech_denoising.html

We highly recommend listening to the results for different approaches.

3. Compare against source separation outside the framework (R2)
We compared with Non-negative Matrix Factorization (NMF) for source separation, and we report results for both unsupervised and supervised NMF:

----unsupervised----
guitar:          SDR: 2.17   SIR: 2.78   SAR: 14.19
congas:        SDR: -0.20  SIR: 0.23   SAR: 14.76
xylophone:  SDR: 2.04   SIR: 3.61   SAR: 12.13

----supervised----
guitar:          SDR: 5.97   SIR: 7.56   SAR: 12.81
congas:        SDR: 1.77  SIR: 2.76   SAR: 11.97
xylophone:  SDR: 8.08   SIR: 12.33   SAR: 11.72

Please let us know for any questions. Thanks again for all the suggestions, which have made this submission stronger.

Best,
Authors.

---

### Public Comment · ~Gabriel_Soares_Xavier1 · 2020-09-11
**Doubt about harmonious convolution.**

Hello, I am a graduate student and I need to better understand this idea of ​​harmonious convolution, there is some material that can explain this more clearly, I would like to implement this in tf.keras.

---

### Public Comment · ~Kazuyoshi_Yoshii2 · 2020-10-20
**HCQT**

Hello. This is an interesting paper in a sense that the harmonic convolution is shown to work as an inductive bias for forming the deep audio prior. However, the idea of the proposed "Harmonic Convolution" is essentially identical to "Harmonic CQT" (HCQT), which has widely been used in the field of music audio processing. Stacking the pitch-shifted versions of an original spectrogram into a tensor enables efficient implementation.
https://github.com/rabitt/ismir2017-deepsalience

---

> ### Author Response · Authors · 2020-10-20
> **Re:HCQT**
>
>
> Thanks for the comments and we'll add related references to our text.
> We would like to point out several major differences:
> 1. CQT-based representations usually has inversion issues, which somehow constraint its usage in generative models, such as ours.
> 2. Our Harmonic convolution is composed of different anchoring and learnable mixtures as well, in addition to sampling at harmonic locations. We showed in our experiments that all of them play an important role in bringing the implicit prior to the network.
> 3. Though we did not experiment with HCQT, but we did experimented with CQT+dilated convolutions, which I would assume is extremely similar to the paper mentioned. However it did not work for us, mostly due to the problems of inverting it back to raw waveform. Also, since the temporal resolution is usually varying for different frequency under CQT, one needs to deform the filter in the temporal axis as well, making sure the convolution window covers the same amount of time.
>
> Hope this could be helpful!
>
> Best,
> Authors.

---

### Decision · Program_Chairs · 2019-12-19

**Decision:**

Accept (Poster)

**Comment:**

This paper introduces a new convolution-like operation, called a Harmonic Convolution (weighted combination of dilated convolutions with different dilation factors/anchors), which operates on the STFT of an audio signal. Experiments are carried on audio denoising tasks and sound separation and seems convincing, but could have been more convincing: (i) with different types of noises for the denoising task (ii) comparison with more methods for sound separation. Apart those two concerns, the authors seem to have addressed most of reviewers' complaints.